# The latitudinal variation in amphibian speciation rates revisited

Adrián García-Rodríguez [1] ✉, Fabricio Villalobos [2], Julián A. Velasco [3], Franz Essl[1] &
Gabriel C. Costa [4]

Speciation can increase regional species richness, however, our knowledge of how and why speciation rates vary across space is still incomplete. Using comprehensive information on species distributions and their phylogenetic relationships, we describe the global spatial patterns of variation in amphibian speciation rates and explore their ecological determinants. We find that overall mean speciation rates in amphibians increase with latitude. This pattern is largely determined by anurans, the most diverse amphibian order. Salamanders, nevertheless, speciate faster in the tropics, whereas caecilians did not exhibit any relationship with latitude. Despite the overall inverse latitudinal trend in mean speciation rates of amphibians, tip-level maximum speciation rates are not necessarily restricted to higher latitudes and can be found in different regions across the globe. Among bioregions, higher mean speciation rates are associated with high past climatic velocity and topographic complexity, both factors potentially promoting isolation, an important primer for speciation. Our results suggest that the emerging inverse pattern of speciation rates in amphibians is likely driven by the combination of a few rapidly speciating clades distributed in higher latitudes and numerous clades in lower latitudes showing heterogeneous rates that, when averaged, pull down the estimations of mean speciation in the areas where they occur.

The increase in species richness from polar to equatorial regions (i.e., the latitudinal diversity gradient - LDG) is a striking biodiversity pattern[1]. The LDG is observed in both terrestrial and marine clades and it has been documented in the fossil record since ~500 Ma[2], with an increased strength over the last 30 Ma[3]. Since it has been first reported around 200 years ago[4], many hypotheses have been proposed to explain the LDG[5], which focus on features that changes across latitudes such as available area, climatic stability, productivity, time elapsed since clade's origin/colonization, and, ultimately, their postulated influence on evolutionary dynamics. However, no widely accepted consensus on the main drivers of this pattern has been reached[5,6].

The LDG is the result of evolutionary and biogeographical processes such as speciation, extinction, and dispersal[7]. Speciation, as the actual process by which new species originate, is one of the major sources determining the variation in species richness at regional and global scales. However, the rate at which speciation events occur can vary across spatial and environmental gradients[8]. Therefore, geographical variation in speciation rates has been hypothesized to play an important role in shaping the LDG[9]. Assuming that speciation is favored by higher productivity and

climatic stability of the tropics, high tropical diversity has been hypothesized to be the outcome of higher speciation rates in the tropics compared to temperate regions[2,9,10]. Nevertheless, evidence shows that latitudinal gradients of species richness and speciation rates are not spatially congruent across several vertebrate groups[11]. For example, birds from tropical and non-tropical regions show similar speciation rates[12], while inverse latitudinal gradients relative to those of species richness have been documented in squamate reptiles and mammals[13,14]. For vascular plants, spatially decoupled patterns of species diversification and species richness have also been recently reported[15].

Different mechanisms may explain why and how higher levels of speciation occur in a region. For instance, species with large range sizes inhabiting larger areas are more prone to experience isolation among populations and divergence leading to allopatric speciation[16,17]; higher resource availability may provide more ecological opportunities leading to increased speciation rates, while total mutations per unit time are higher in warmer regions accelerating the speciation process[5,18]. In addition, historically climatically stable regions may decrease extinction rates, and favor

[1]Division of BioInvasions, Global Change & Macroecology, Department of Botany and Biodiversity Research, University of Vienna, Rennweg 14, Vienna, 1030, Austria. [2]Red de Biología Evolutiva, Instituto de Ecología A.C., Xalapa, Veracruz, CP, 91073, Mexico. [3]Instituto de Ciencias de la Atmósfera y Cambio Climático, Universidad Nacional Autónoma de México, Ciudad Universitaria, Mexico City, CP, 04510, Mexico. [4]Department of Biology and Environmental Sciences, Auburn University at Montgomery, Montgomery, AL, 36117, USA. ✉e-mail: adrian.garcia@univie.ac.at

specialization to narrower niches, a condition likely linked to higher speciation rates[5]. From an ecological perspective, it has been proposed that the intensity of species interactions may influence speciation rates, where higher rates of speciation are expected in areas with stronger ecological interactions, for example, as an outcome of interspecific competition[19–21]. Finally, considering that speciation frequently occurs in allopatry, topography-driven isolation has been hypothesized to be associated with increases in speciation rates[22,23].

Despite its potential importance in shaping biodiversity patterns, geographic variation of speciation rates and its potential drivers are not well understood[6,8]. Many LDG studies have focused on the effects of environmental predictors on current patterns of species richness[24–26], but few have explored the link between those predictors and the rate at which species arise[8]. To better understand the LDG, it is necessary to first evaluate the underlying causes of speciation gradients, and then, assess how these processes influence current geographic biodiversity patterns[6]. The increasing availability of near-complete species-level phylogenies, environmental data, and geographic distribution ranges can be used to reveal how these forces underpin the geographic variation in speciation rates and its actual contribution to richness patterns[6,27]. Indeed, steps towards filling this gap have been made already for most terrestrial vertebrate clades (birds, mammals, and squamate reptiles) with the notable exception of amphibians.

Here, we aim to evaluate competing hypotheses on the drivers of speciation gradients in amphibians. We combined a near-complete phylogeny of the group[28] current and historic environmental data (i.e., paleo- and current climate, topography), and data on species distribution. We used this data to describe the latitudinal distribution of amphibian speciation rates, both for the entire amphibian radiation and for each order independently. Next, we assessed which predictors better explain the latitudinal patterns of speciation rates in this vertebrate class by testing six non-mutually exclusive hypotheses (Table 1).

## Results
### Global patterns of amphibian speciation
Our results showed a trend of moderate, yet statistically significant higher speciation rates at non-tropical regions when looking at the pattern for all amphibians (Fig. 1). This same pattern emerged when assessing anurans. In the case of salamanders, we found the opposite pattern; peaks of speciation are located in Mesoamerica and steeply decrease towards temperate latitudes. For caecilians no clear pattern was detected (Fig. 1; Table 2).

Considering all species, we found that within grid cells maximum speciation rates, are not necessarily restricted to higher latitudes but occur in different regions across the globe, including eastern USA, Mesoamerica, the Amazon, the Atlantic Rainforest, Southern Andes, Madagascar, and Southeast Asia (Fig. 2). We found that in these regions the speciation rates are very heterogeneous, as inferred from the high grid cell level coefficient of variation in this metric, which in most cases exceeds 30% (Fig. 2). In contrast, we identified some of the most homogeneous grid cells in terms of speciation rates in the northern portions of both the Nearctic and Palearctic regions. In some of these regions, high means of speciation rates are determined by species-poor assemblages (Fig. S4).

### Tropical versus non-tropical speciation rates
We split our dataset based on the species' latitudinal midpoints and classified 5565 species as tropical and 1332 species as non-tropical, nearly 80% and 20% of the studied species, respectively. The FISSE model confirms that speciation rates tend to be slightly higher in the non-tropical regions when looking at overall amphibian diversity (mean lambda $_{non-tropical}$ = 0.089; mean lambda $_{tropical}$ = 0.083. Within orders, this overall trend is mirrored by frogs (mean lambda $_{non-tropical}$ = 0.092; mean lambda $_{tropical}$ = 0.082) but is the opposite for caecilians (mean lambda $_{non-tropical}$ = 0.032; mean lambda $_{tropical}$ = 0.039), and even more so for salamanders (mean lambda $_{non-tropical}$ = 0.078; mean lambda $_{tropical}$ = 0.121) (Fig. 3).

Among bioregions, we found the fastest estimations of mean speciation rates in Temperate South America (mean speciation = 0.080), Boreal North

America (mean speciation = 0.078), and the North American Grasslands (mean speciation = 0.077). Instead, mean speciation rates were lower in Dry Forest Madagascar (mean speciation = 0.0485) and the Mediterranean Afrotropics (mean speciation = 0.0487).

### Drivers of variation in amphibian speciation rates
Among bioregions, the Deserts of Eurasia had the largest area, whereas the Mediterranean Afrotropics was the smallest bioregion included in the study (Supplementary Table 1). Productivity reached the highest levels in the Moist Forests of South America and Madagascar, whereas the North American Tundra and the Deserts of Eurasia had the lowest productivity (Supplementary Table 1). Temperature was lowest in the Tundras and Boreal regions and higher in the Dry Forests of Australia and Madagascar (Supplementary Table 1). Australian biomes had the lowest topographic complexity, whereas roughness peaked in Temperate South America. Regarding climate change velocity, we found the highest values in the Tropical Moist Forest of South America, the Grasslands of Eurasia, and the boreal areas of North America and Eurasia. (Supplementary Table 1). We found the highest values of the Net Relatedness Index in the Afrotropical Moist Forests, the Indo-Malay Dry Forest, and the Temperate regions of South America (Supplementary Table 1).

Among the six variables evaluated, we found that average climatic velocity and topographic complexity are both significantly and positively correlated with the variation in amphibian speciation rates across bioregions (Fig. 4). Accordingly, we found that speciation rates are higher in regions with rougher topographies, and where climatic oscillations have occurred at a higher pace in the last 3 million years (Fig. 4, Table 3).

## Discussion
Given the critical role of the speciation process in generating diversity, a key question for biogeographers and evolutionary biologists is how speciation rates vary across space[6,8]. When analyzing the entire amphibian radiation, we found an inverse latitudinal gradient of speciation, relative to the classical gradient of increasing species richness towards the Tropics (Supplementary Fig. 1). Nevertheless, our results show heterogeneous latitudinal patterns of speciation rates among amphibian orders. Anurans mostly drive an overall increase in amphibian mean speciation rate with latitude, whereas salamanders speciate faster in lower latitudes and caecilians did not show a latitudinal pattern of speciation. Among the drivers tested, we found significant correlations of mean speciation rates with topographic complexity, and climate change velocity. Overall, speciation seems to be boosted in regions of rough terrain and to a minor extent in regions that have experienced stronger climatic instability over the last 3 million years.

The spatial variation in speciation rates has been pointed out as one major factor shaping the distribution of species richness across the globe[10,29,30]. Although most taxonomic groups, including amphibians[31,32], follow a classic LDG with higher richness in tropical regions, inverse latitudinal gradients of speciation, such as the one found here for the class Amphibia, have also been documented recently for other taxonomic groups. Growing evidence have documented for example that lineages of plants[33]; squamate reptiles[13]; marine fishes[34] and mammals[14] do not speciate faster in the tropics where species richness is highest. This consistent pattern has recently challenged conventional theory linking variation in speciation rates and latitudinal gradients of species richness. Perhaps, the most popular of these is the Metabolic Theory of Ecology, which states that high ambient energy (i.e. the tropics) can increase speciation rates, through increases in metabolic and mutation rates[35,36]. Still, based on increasing evidence of higher speciation rates in temperate regions recent studies decisively discarded a mechanistic link between diversification rates and species richness[15].

In amphibians, we also found an inverse latitudinal speciation gradient relative to species richness. Nevertheless, we also noted that speciation rates may vary highly within species assemblages in some regions of the world. This may have an important -yet underestimated- impact on calculating mean speciation values, on which descriptions of latitudinal gradients are

**Table 1 | Potential drivers of variation in amphibian speciation rates tested in this study**

| Predictor | Mechanism | References | Predictions about latitude | Expected correlations |
|---|---|---|---|---|
| Time-integrated Area | Larger areas can support species with larger geographic ranges. Given enough time, large areas increase the potential for population isolation, leading to higher speciation. | 16–18 | Tropical regions have been historically larger providing sufficient area and enough time for speciation to occur more often in the tropics | |
| Productivity | In regions of high productivity, the availability of more and heterogeneous resources can support larger populations and increase ecological opportunities for speciation. | 28–30 | Tropical regions have been more productive, supporting increased numbers of individuals and species over time, which may lead to higher speciation rates in the tropics. | |
| Temperature | Mutation rates are faster, and generation times are shorter at higher temperatures. These processes could promote reproductive isolation, increasing speciation rates. | 31–33 | The warmer tropical temperatures provide the conditions for increased rates of speciation in the tropics | |
| Climatic stability | Organisms from climatically stable regions have sufficient time for divergent selection and speciation to operate. Milder climatic oscillations favor speciation. | 34,35 | Milder climatic oscillations in the Tropics provide sufficient time to promote higher tropical speciation rates | |
| Biotic interactions | Higher biotic interactions increase competition and predation. These pressures may result in higher speciation rates via adaptation or decreased speciation due to niche-filling | 36–38 | Higher interactions in the Tropics promote speciation. Higher niche filling in the tropics decreases speciation rates | |
| Topographic complexity | Isolation and climatic heterogeneity in mountains serve as barriers to gene flow, increasing the chances of speciation. | 23,39,40 | Stronger allopatric effect of Tropical mountains favor higher speciation at such latitudes. | |

We provide the rationale of the proposed mechanism involved, existing evidence, predictions, and the expected pattern. The plots show the expected correlations between speciation rates (solid lines) and each predictor variable (dashed lines) across different latitudes.

often based. Our results show that the effect of co-occurring fast and slow speciating taxa is widespread in species-rich assemblages, such as those in tropical regions (e.g. Isthmian Central America, Madagascar, the Amazon basin, or Southeast Asia) or regional hotspots of amphibian diversity in other latitudes (e.g. southeastern USA[37],). For example, in the Mesoamerican hotspot the distribution of the fast-speciating tropical Bolitoglossine salamanders overlaps with other species showing very slow rates of speciation, such as the Mexican burrowing toad, *Rhynophrynus dorsalis*. In many cases, these co-occurrence patterns translate into intermediate mean speciation rates in species rich assemblages at lower latitudes, contrary to several high latitude regions, with less diverse amphibian faunas but with estimated speciation rates above the average. For example, frogs of the genus *Alsodes* and plethodontid salamanders of various genera (e.g. *Eurycea and Plethodon*) are distributed out of the Tropics, respectively, and show some of the fastest speciation rates among the over 7200 species studied. Certainly, the co-occurrence of fast and slow speciating lineages in some regions reveals the importance of accounting for the idiosyncratic responses of lineages to shared eco-evolutionary pressures, and which could lead to fast/slow speciation in some groups but not in others. Likewise, other sources of rate heterogeneity such as the variation in speciation rates among groups diversifying at different times should be better incorporated to fully understand the temporal dynamics that result in traditional – following species richness – as well as inverse – opposite to species richness – latitudinal gradients of speciation[38].

We consider that ignoring the differential contribution of each clade to the overall gradient of speciation may lead to an underestimation of the actual role of the tropics in generating biodiversity. For example, deconstructing the geographic patterns of speciation at the order level, we

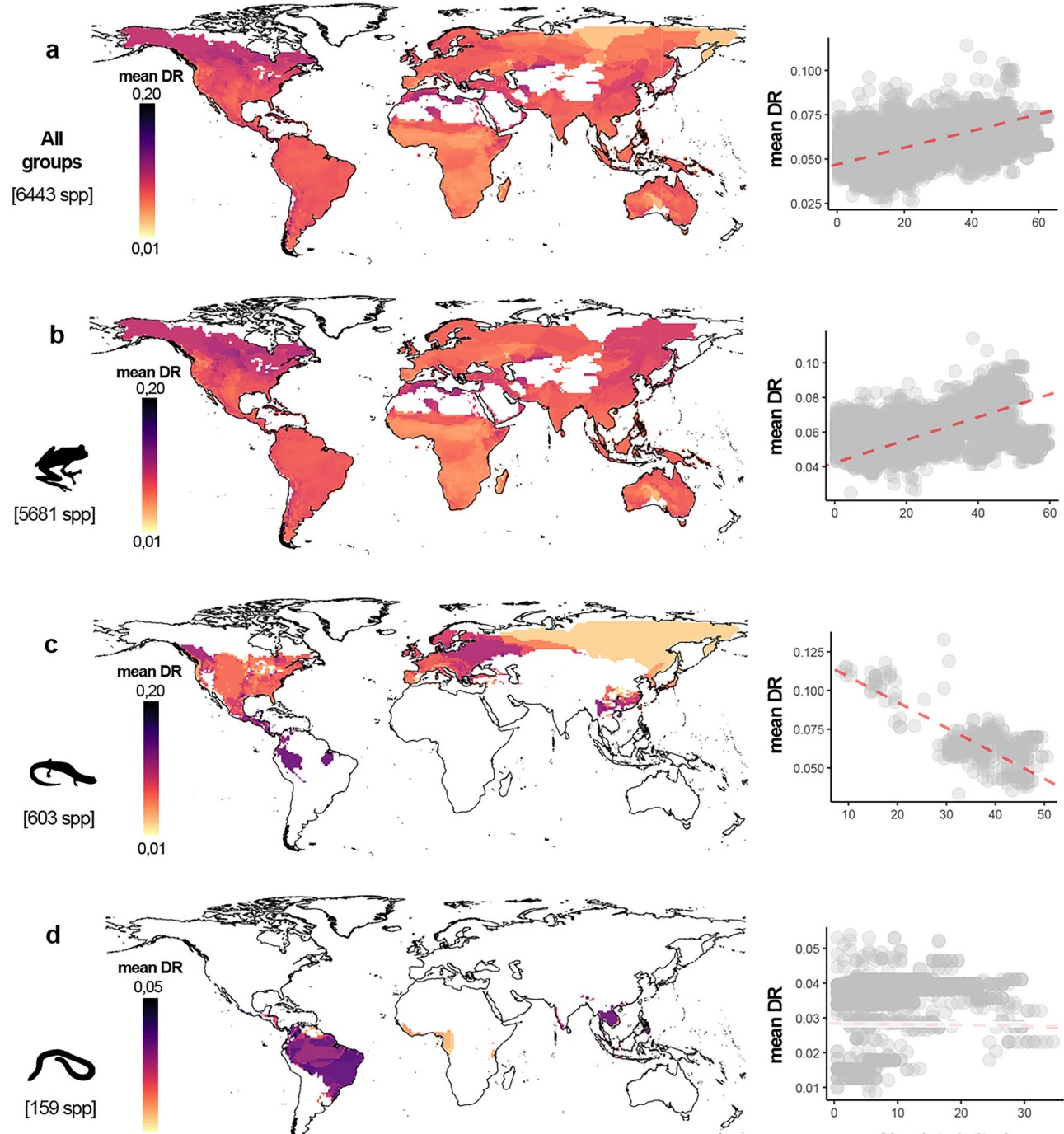

**Fig. 1 | Global geographic patterns of speciation rates in amphibians.** Speciation variation in (**a**) the entire amphibian radiation and within the orders Anura (**b**), Caudata (**c**), and Gymnophiona (**d**). Maps show the geographic distribution of mean speciation rates estimated from tip-level DRs and averaged for each grid cell based on local species assemblages. Plots on the right-hand side show results of SAR's models evaluating the relationship between mean DR and absolute latitude. Regressions show a trend of increasing speciation rates at higher latitudes, both in the full amphibian radiation (**a**) and in the case of frogs and toads (**b**). The pattern is opposite for salamanders (**c**) where speciation rates increase towards the tropics. For caecilians, no clear pattern of increase or decrease with latitude was (**d**). The silhouettes used here and in Fig. 3 were obtained from www.phylopic.org/ (*Rhacophorus lateralis* by Vijay Karthick, *Pseudoeurycea amuzga* and *Dermophis mexicanus* by Jose Carlos Arenas-Monroy).

found that anurans also follow the overall inverse latitudinal gradient of speciation described here for amphibians. This is to be expected as frogs and toads account for over 85% of the total amphibian species in the world[39]. Nevertheless, it has been demonstrated that peaks of speciation in frogs and toads occur both in families with mostly tropical distributions (e.g., Centrolenidae or Phyllomedusidae) and families reaching temperate regions (e.g., Alsodidae or Bufonidae)[23]. As such, differences in average speciation in Anura across the globe may emerge from the high diversity of families and their heterogeneous speciation rates in the tropics that contrast with the few, but fast speciating, lineages occurring in non-tropical regions.

For salamanders (Caudata, 767 species[39];), the results contrast with the overall trend, with faster speciation rates occurring in the tropics. Interestingly, this also represents an inverse speciation gradient relative to the

**Table 2 | Results of the Spatial Autoregressive (SAR) linear models implemented to assess the relationship between speciation rate and absolute latitude, for all amphibian species and at the order level**

| Taxa | Estimate | Std. error | *p*-value | Adjusted R² | Pseudo-R² |
|---|---|---|---|---|---|
| All amphibians | 4.70 e-04 | 6.50 e-05 | <0.05 | 0.159 | 0.833 |
| Anura | 6.59 e-04 | 6.75 e-05 | <0.05 | 0.158 | 0.896 |
| Caudata | −1.64 e-03 | 2.71 e-04 | <0.05 | 0.531 | 0.827 |
| Gymnophiona | −2.59 e-05 | 1.37 e-4 | <0.05 | 0.048 | 0.778 |

For each model the regression coefficients are shown along with the significance level, and a comparison between the R² from an Ordinary Least Square regression and the Nagelkerke pseudo R² once accounting for spatial autocorrelation with the SAR models.

**Fig. 2 | Global patterns of within-assemblage maximum speciation rates and speciation rate variability for amphibians at a 1 × 1° resolution. a** Map showing the distribution of fastest speciation rates estimated for each grid cell. **b** Speciation rate heterogeneity inferred from the coefficient of variation among the tip-level rates computed for all the species constituting each assemblage.

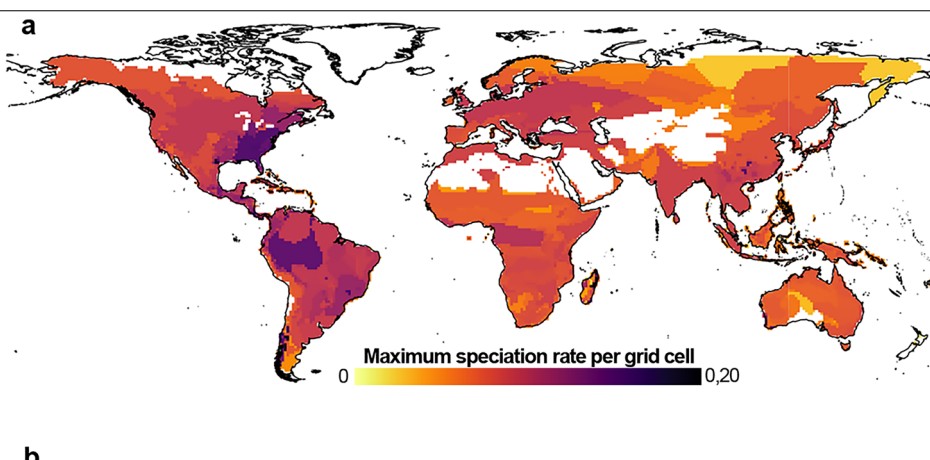

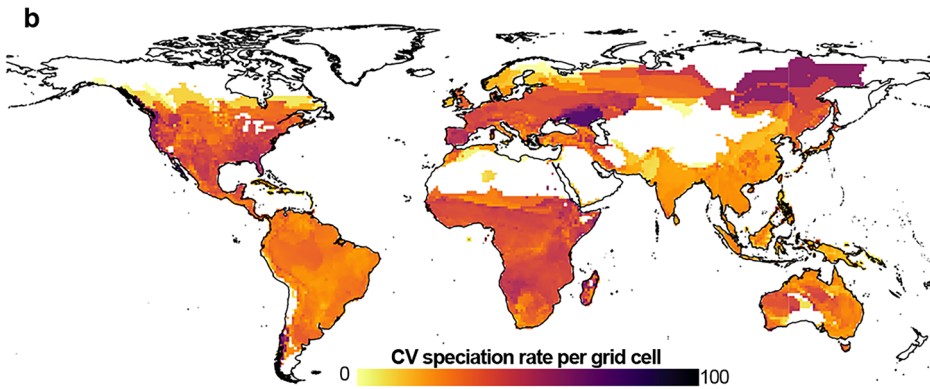

known species richness in the group[40]. Nearly 40% of the total diversity of Caudata[39] is in the bolitoglossine salamanders from the Plethodontidae family. This family has a temperate origin and bolitoglossines are the only lineage that successfully dispersed into the tropics[41]. Moreover, this group of Plethodontid salamanders radiated more recently in the mountainous regions of Mexico and Central America[41–44]. Such recent speciation events in more tropical latitudes by the major lineage of Caudata explain the faster and more homogeneous speciation rates that we found for the entire order near the tropics.

For caecilians, whose distribution is mostly restricted to the tropics[45], we did not find any clear trend regarding latitude when applying an assemblage-based approach. This could be the result of high variability of speciation rates observed for the group near the Equator. Consistently, in our complementary analysis at the species level, we did not find major differences in speciation between tropical and non-tropical caecilians (Fig. 3). Overall, tip speciation rates are lower for caecilians in comparison to other amphibians (Fig. 2). This confirms previous findings suggesting that the specialized and evolutionary conserved ecology of caecilians along with their low dispersal capacity may have constrained their diversification[46–48], in contrast to other groups where reduced dispersal has been linked to

increased speciation rates[49,50]. The combination of this particular speciation dynamics in caecilians and their distribution mostly restricted to low latitudes, certainly contributes to the overall speciation rate heterogeneity found in some neotropical regions, where fast and slow speciation groups co-occur.

Regarding drivers of speciation, we found that the rate at which species originate increases in biomes with faster climatic velocity and high topographic complexity. Mountainous regions have been long recognized as hotspots of biodiversity and endemism[51,52] and therefore, considered engines of diversification[53–55]. On one hand, the rough terrains of mountains may impose physical barriers for dispersal, increasing isolation that may eventually lead to allopatric speciation[56–58]. At the same time wide elevational ranges cover large gradients of climatic variation at short distances, providing a mosaic of conditions that can drive adaptive divergence and may end up in the formation of new species[59–61]. For amphibians, it has been demonstrated that this dual influence results in significant increases in the speciation rates of taxa distributed in regions of high topographic complexity[23,62], a pattern also mirrored by our results.

Long-term regional climatic stability has often been proposed as a driver of biodiversity by potentially reducing extinction rates and allowing

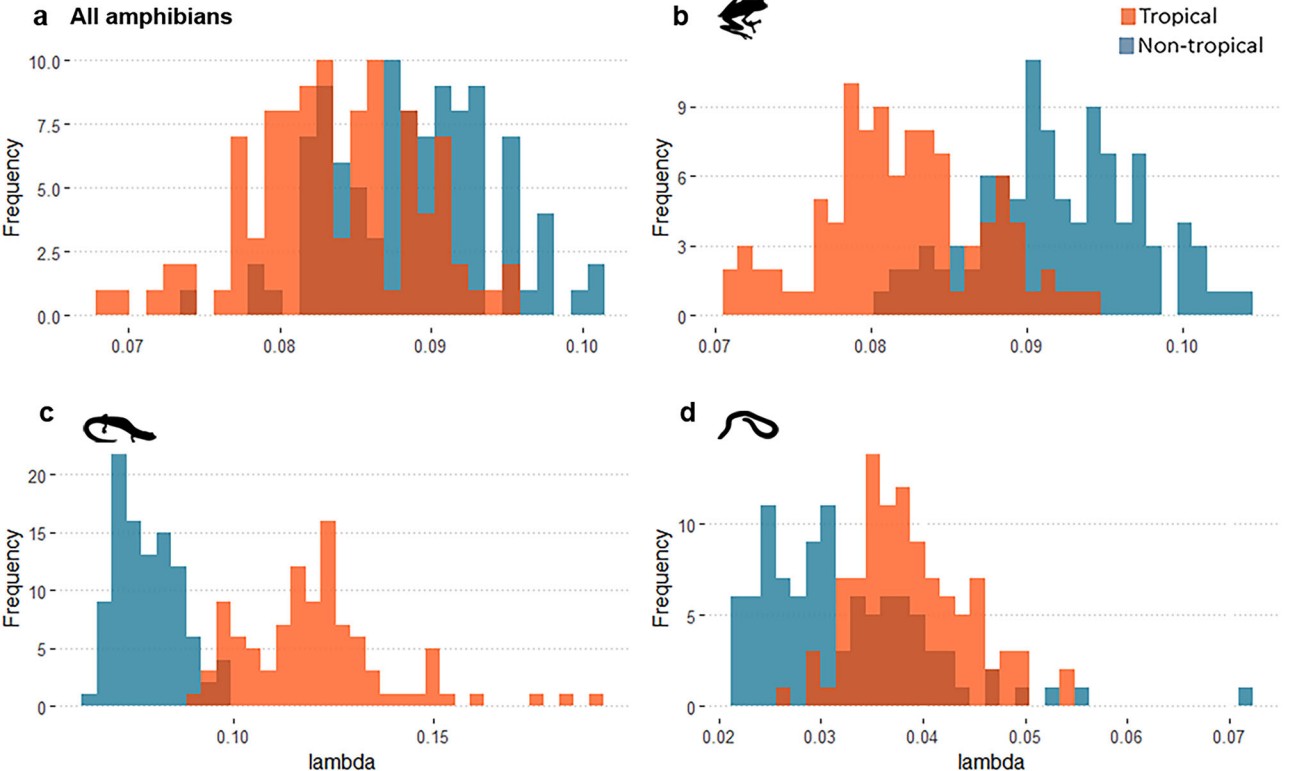

**Fig. 3 | Binary state trait-dependent speciation estimated from FiSSE analyses.** Histograms show the distribution of speciation rates for tropical (orange) and non-tropical species (blue) estimated from 100 random trees. Each panel represents a different taxonomic grouping: (**a**) all amphibians, (**b**). anurans, (**c**). salamanders, and (**d**). caecilians. Dashed vertical lines denote the mean values of lambda obtained for each group (red for tropical; light blue for non-tropical) from the 100 random trees analyzed.

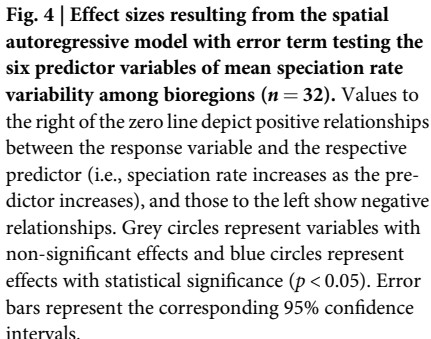

**Fig. 4 | Effect sizes resulting from the spatial autoregressive model with error term testing the six predictor variables of mean speciation rate variability among bioregions ($n = 32$).** Values to the right of the zero line depict positive relationships between the response variable and the respective predictor (i.e., speciation rate increases as the predictor increases), and those to the left show negative relationships. Grey circles represent variables with non-significant effects and blue circles represent effects with statistical significance ($p < 0.05$). Error bars represent the corresponding 95% confidence intervals.

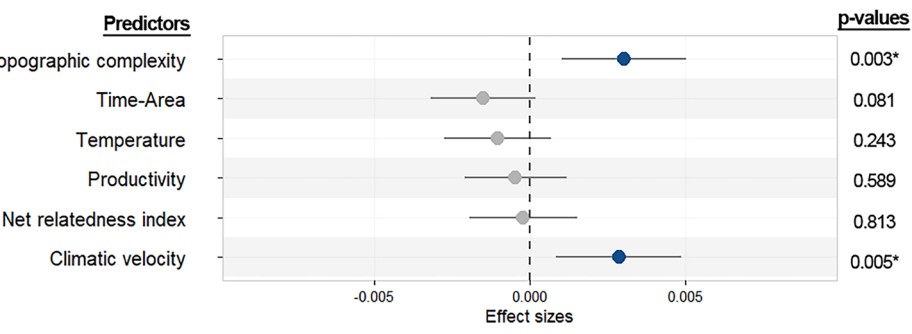

taxa to persist in stable environments[2]. Stability might also enable time for morphological differentiation and promote specialization that could, in theory, enhance speciation rates[5]. However, our results reveal an increase in mean speciation rates in regions that have experienced greater climatic instability over the past ~3.3 million years (Mya). This aligns with findings in other groups, such as mammals, where higher speciation rates are linked to regions of climatic instability, measured as the difference in mean annual temperature between the present and the last glacial maximum[14], a more recent timeframe than in our study. Given the evolutionary timeline of amphibians (ca. 300 Mya), we chose to use the longest available climatic dataset to capture more extended periods of environmental change. While we recognize that only a portion of the studied species are products of speciation within the last ~3Mya, these results suggest that regions of climatic instability may indeed foster speciation in amphibians, despite their deep evolutionary history. We stress that the ability of past climatic dynamics to inform about processes on much deeper evolutionary scales

must be interpreted cautiously and remain an issue to be improved in future studies.

Recently, it has been proposed that inverse latitudinal patterns of speciation could be determined by the hitherto underappreciated latitudinal taxonomy gradient (LTG)[63]. This phenomenon is hypothesized to result from a taxonomic debt at low latitudes owing to geographic biases in the application of species definitions[63–65]. This taxonomic debt seems to be higher in the tropics, in part due to the disproportionate number of cryptic species occurring in those regions as recently shown for birds[63]. A similar situation could be expected for amphibians, a group with a particularly unstable taxonomy during the last decades mainly due to the continuous discovery of new species, many of them masked within cryptic complexes[66,67]. After a brief review assessing geographic and taxonomic biases in the species not included in the phylogeny here used, we found that among the c. 1400 amphibian species described in the past decade, a significant portion are tropical, with nearly 50% clustered within just five of 52

**Table 3 | Resulting coefficients from the spatial autoregressive regression model testing relationships at the bioregion level among mean speciation and the six predictors evaluated**

| Predictor | Estimate | Std. Error | z value | *p*-value |
|---|---|---|---|---|
| Time-area | −1.49 e-03 | 8.60 e-04 | −1.74 | 0.081 |
| Productivity | −4.53 e-04 | 8.41 e-04 | −0.54 | 0.589 |
| Temperature | −1.02 e-03 | 8.77 e-04 | −1.16 | 0.243 |
| Topographic complexity | 3.03 e-03 | 1.02 e-03 | 2.96 | 0.003 |
| Net Relatedness Index | −2.08 e-04 | 8.83 e-04 | −0.24 | 0.813 |
| Climatic velocity | 2.87 e-03 | 1.03 e-03 | 2.79 | 0.005 |

Nagelkerke pseudo $R^2$ = 0.79; *p*-value: 2.83e-09

families recently represented. Although incorporating proposed solutions to address these biases is challenging[63]. acknowledging them is important due to their potential impact on speciation rate estimates[68]. Notably, the exclusion of numerous tropical lineages may lead to an underestimation of dynamic diversification in these regions. Interestingly, the accelerated rates of species description in this group provide a good opportunity to further quantify the impact of a dynamic LTG on speciation estimates in specific clades and regions. Finally, our chosen speciation rate metric has been criticized as performing poorly in some scenarios compared to model-based metrics such as those derived from BAMM or ClaDS[69]. However, our exploratory analyses showed that tip-level (species) and assemblage-level values of speciation rate are highly correlated between these three metrics ensuring that the overall patterns won't change depending on the metric as, on average, species/assemblages with higher (or slower) rates for one metric would still present higher (or lower) rates with any other metric; especially in the case of grid-cell level where the value is summarized across species.

Our findings add further evidence to the occurrence of inverse latitudinal gradients in speciation rates recently reported also for other taxonomic groups. However, unlike ours, few studies have explicitly explored the underlying causes of such patterns. From our analysis, climatic instability and high topographic complexity emerge as drivers boosting amphibian speciation. Nevertheless, the role of historical climatic dynamics should be explored in more detail evaluating whether available paleoclimate reconstructions represent appropriate timeframes for capturing lineage diversification. Importantly, while our findings suggest that speciation dynamics is decoupled from the contemporary patterns of amphibian species richness, our data also shows that some of the maximum values of speciation occur in many speciose tropical and subtropical regions. We hypothesize that local speciation rate heterogeneity is likely influenced by species richness and therefore has a particularly strong impact on the estimation of the average speciation rates in the tropics, obscuring the actual relevance of these regions as biodiversity cradles.

## Methods
### Estimating amphibian speciation rates
We estimated tip-level speciation rates across the most complete amphibian phylogeny containing 7238 species[28], more than 80% of the known extant amphibian diversity. We used the Diversification Rate metric (DR)[70] as a proxy for speciation rates. This metric derives from a non-model-based approach to estimate macroevolutionary tip rates based on the number of splitting events and the internode distances along the root-to-tip path of the phylogeny while giving greater weight to branches closer to the present[70]. Although DR was originally proposed as a statistic to estimate species-level lineage diversification rate, recent work has supported this metric as a better estimator of speciation instead of net diversification, with subsequent exploration demonstrating that DR is particularly informative for speciation rates[71–73]. Empirical tests providing evidence that DR is highly correlated with speciation rates[71], also suggest that this metric can be computationally more efficient when dealing with large phylogenies than model-based metrics, like those resulting from the Bayesian Analysis of

Macroevolutionary Mixtures (BAMM[74,75]) or the Cladogenetic Diversification rate Shift model (ClaDS[76]). To incorporate topological uncertainty in our estimates we calculated the DR metric from 100 random full trees (i.e., containing all 7238 species available) of the posterior distribution of phylogenetic trees from ref. 28. We explored the consistency of our estimates with those of model-based metrics by implementing BAMM and ClaDS analyses, but only on a smaller set of ten random trees. Given the strong correlation between the average speciation rates obtained with the three approaches, for both species and assemblages, ($r > 0.7$; see Supplementary Fig. 2), we opted to use DR for the subsequent analyses to better account for topological uncertainty using 100 instead of ten trees.

### Global geographic patterns of amphibian speciation rates
To analyze the geographic patterns of amphibian speciation rates, we first adopted an approach based on grid-cells. We used the R package LetsR[77] to create a 1-degree spatial resolution (~100 x 100 km near the Equator) presence-absence matrix for species in the phylogeny having available range maps in the dataset of Digital Distribution Maps of the IUCN Red List of Threatened Species (www.iucnredlist.org). After excluding species introduced outside their native ranges, our dataset consisted of 6443 species, representing ~72% of the 8869 currently described species[39]. We further estimated speciation rates per grid cell by averaging the species-specific speciation rates, as the arithmetic mean of the species co-occurring within each grid cell.

Some authors have argued that dynamic species ranges may mask potential relationships between current distributions and the geography of speciation[78]. However, strong evidence supporting that a signal of speciation can be recovered regardless of range dynamics is well-established in the literature for a variety of organisms, from fossil mollusks to living insects and mammals[79–81]. Accordingly, we argue that geographical signals of speciation are still discernable today. Further, most amphibian species have a poor dispersal ability[82] and are adapted to specific environmental conditions, resulting in a high proportion of species with small range sizes[83–85]. Therefore, we assumed that the effects of range dynamics on the geographical signal of interest should be moderate at our studied scales and resolution.

We deconstructed the observed pattern for the entire class by repeating the previous procedure with the subsets of known species belonging to each amphibian order: Anura, Caudata, and Gymnophiona. For these analyses, we considered 5681 frogs and toads (~73% of all anurans); 603 salamander species (~73% of all Caudata) and 159 caecilian species (~71% of all Gymnophiona). To map speciation patterns, we rasterized the presence-absence matrix of each group and plotted the global distribution patterns of mean speciation rates per one-degree cell. Moreover, we created maps to visualize the distribution of maximum speciation rates estimated for each grid-cell and the distribution of speciation rate heterogeneity. For the latter, we calculated the coefficient of variation of speciation rates within grid-cells, a proxy of heterogeneity that is not prone to be influenced by differences in species richness among grid cells.

It has been shown that diversification rates are time-dependent[86] which could add noise to this analysis if it were the case that species' ages are latitudinally structured. Thus, we tested the relationship between speciation rates and species ages (derived from their branch length), which indeed resulted in a significantly positive correlation (Supplementary Fig. 3). Nevertheless, we also found that the geographic distribution of ages is not particularly biased to specific latitudes (Supplementary Fig. 3). Therefore, we consider that such time-dependency of the estimated speciation rates does not have a strong effect on the geographic patterns that we observed.

To test the relationship between mean speciation rates per one-degree resolution cell and the respective absolute latitude at a global scale, we fitted spatial autoregressive models (SAR) with a spatial error term to account for spatial autocorrelation. These models use weight matrices that specify the strength of interaction between neighboring sites to account for spatial autocorrelation[87,88]. We created a neighborhood matrix for each model using five degrees as the maximum connectivity distance, and an inverse square distance weighting function ($1/d2$). To estimate the variance

explained by each model, we calculated pseudo-R2 based on the Nagelkerke formula[89]. Pseudo-R$^2$ represents the variance explained by both the predictor variable and space. To include the variance explained by the predictor alone, we reported the R$^2$ from least-squares linear models.

## Speciation in tropical and non-tropical groups

Additionally, we tested if speciation rates between tropical and non-tropical species differ. For this, we focused on species (instead of grid-cells) and used the latitudinal centroid of each species range, extracted with LetsR[77] to assign the species geographic region. We classified as tropical those species with latitudinal mid-points between the Tropics of Cancer (23.44°N) and Capricorn (23.44°S) and as non-tropical all the remaining, as implemented in previous studies[90]. Based on this discrete latitudinal classification, we ran 100 FiSSE tests (Fast, intuitive, state-dependent speciation-extinction model[91], over an equal number of random trees from the posterior distribution. FiSSE is a non-parametric approach proposed to identify state-dependent diversification in binary traits, by inspecting the distribution of branch lengths of each state (here the tropical versus non-tropical distribution) without assuming an underlying model structure[91].

## Potential drivers of spatial gradients in amphibian speciation rates

We analysed which drivers better predict the spatial variation of speciation rates in amphibians and used these findings to test non-mutually exclusive hypotheses (Table 1). For this analysis we used as study units 32 bioregions defined by ref. 92 which are nested within the world's main biogeographic realms according to ref. 93. For each bioregion, we compiled mean speciation rate as the response variable and estimations of time-integrated area, area productivity, temperature, climatic instability, intra-class competition, and topographic complexity as predictors.

For time-integrated area, area productivity, and temperature we used estimations made by[92] for each bioregion. The time-integrated area aims to capture the variable available area for each bioregion over a 55 million years' period by summing the area estimated for each bioregion in one-million-year time slices. Area productivity is the summed productivity that results from the product of average productivity estimated from 17 different global models of terrestrial net primary productivity[94] and the size of the bioregion. Temperature for each bioregion was derived from average annual temperatures from the gridded climatology 1961–1990 made available by the University of East Anglia's Climatic Research Unit[95]. Further details on these three variables are available in[92].

We calculated the velocity of historic climate change through time using the 11 time-horizons available in the Paleoclim database[96] which encompass different past climate change abrupt events for the last 3 mya (e.g., the transition between Pleistocene-Holocene known as the Younger Dryas Stadial[97]; and current conditions using a set of 14 bioclimatic variables from the WorldClim database[98]. This metric captures the historic climate-change velocity for a combination of temperature and precipitation variables. Climate-change velocity represents a local spatial shift index of how local climate change occurs across space (km) and time (years)[99]. We followed[99] and calculated this metric for each period vs. current conditions. Next, we calculated the average of climate-change velocity for all periods.

To infer the potential degree of biotic interactions within each bioregion, we estimated the Net Relatedness Index (NRI[100];) as a proxy of competition. Considering that competitive exclusion is expected to be higher among closely related species[100], we estimated the NRI which measures average branch lengths separating taxa within communities[101]. Although this metric is not strictly linked to competitive exclusion[102,103], it can provide insights into these interactions under the assumption that competition tends to be stronger among close relatives[100,102,104]. We estimated the Net Relatedness Index (NRI) for each grid cell considering a regional species pool for each bioregion (i.e., including all species present at each bioregion) and randomizing their phylogenetic relationships while keeping the observed species richness pattern and the species' range-size frequency distribution.

To estimate topographic complexity, we calculated the terrain's average roughness for each bioregion using cells from an elevation layer of 5 km resolution downloaded from www.worldclim.org. This metric returns the difference between the maximum and the minimum value of a cell and its eight surrounding cells[105]. Next, we averaged all cells within each bioregion to obtain a mean value of roughness per study region. For all subsequent analyses, we corroborated that variables were not collinear (Supplementary Fig. 4). We tested for significant relationships among these predictor variables and the variation in mean speciation rates across bioregions by running additional SAR models with all predictors to account for spatial autocorrelation. Given the large areas covered by our study units, we determined maximum neighbor distances by fitting two sets of lag and error models, with maximum connectivity distances ranging from 10 to 100 degrees. From these models, we selected 40 degrees as the optimal distance based on the lowest Akaike Information Criterion (AIC) value among all models. To estimate variance explained by the predictors while accounting for spatial structure, we calculated pseudo-R2 based on the Nagelkerke formula[89].

## Reporting summary

Further information on research design is available in the Nature Portfolio Reporting Summary linked to this article.

## Data availability

Phylogenetic trees used to estimate speciation rates and Net Relatedness Index are available at https://vertlife.org/files_20170703/#amphibians by Jetz and Pyron (2018). Topographic complexity index was calculated using a 30 s resolution elevation layer available at https://www.worldclim.org/. Species range polygons used here are available at: https://www.iucnredlist.org/. Past climatic data was downloaded from http://www.paleoclim.org/.

## Code availability

Codes and final datasets needed to reproduce the analyses conducted in this study are available at https://github.com/Garcia-Rodriguez/Amph-SpecGradients and have been also deposited in Zenodo[106].

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

## Acknowledgements
A.G.R. and F.E. appreciate funding from Austrian Science Fund FWF (grant DOI 10.55776/ESP5590324 and grant no. I 5825-B, respectively). F.V. thanks INECOL and CONAHCYT (CB project #A1-S-34563) for their support. J.A.V. is grateful to DGAPA for support through the grant UNAM-DGAPA-PAPIIT IA206523. We thank the insightful comments from two anonymous reviewers that helped improve the final version of our manuscript. This research was funded in whole or in part by the Austrian Science Fund (FWF) [grant DOI 10.55776/ESP5590324, available via https://www.fwf.ac.at/en/discover/research-radar]. For open access purposes, the author has applied a CC BY public copyright license to any author accepted manuscript version arising from this submission.

## Author contributions
A.G.R., G.C., F.V. and J.A.V. conceptualized the research. A.G.R. and J.A.V. analyzed the data. A.G.R. led the writing with significant input from F.V., J.A.V., F.E. and G.C. who contributed to preliminary versions and approved the final version of the manuscript.

## Competing interests
The authors declare no competing interests.
