## [Transparent Peer Review file · Communications Biology]

The latitudinal variation in amphibian speciation rates revisited

Corresponding Author: Dr Adrián García-Rodríguez

Version 0:

Reviewer comments:

Reviewer #1

(Remarks to the Author)

The authors provide an analysis of global speciation rates in amphibians—demonstrating that the latitudinal diversity gradient is not simply explained by higher speciation rates in the tropics. In fact, speciation rates (overall, which was driven by frogs) are higher at higher absolute latitudes, and the authors provide some additional analyses to attempt to disentangle potential explanations. Overall, I found the study to be well-conceived and the manuscript to be well-written and clear. I think that it's a valuable contribution to the field. Below, I provide some comments—the most important of which I believe to be related to the implications of the way the authors calculated heterogeneity. Barring that, I have no major concerns or suggestions for improvement.

Lines 11–12: Where exactly does the claim that "we detected peaks of speciation in both low- and high-latitude taxa" come from? To me, this implied that the authors would document and discuss taxon-specific speciation rates somewhere in the paper, but I couldn't find these data in the results or supplemental materials (although there is some discussion of the bolitoglossine plethodontids from Lines 329–335). If this is not what they mean to imply, perhaps this could be rephrased for clarity or removed from the abstract, where I think it is given greater emphasis than elsewhere in the manuscript.

Line 40: A minor point here. At various points, the authors either do or don't capitalize "tropics" (and "tropical"). I encourage them to format this consistently throughout the manuscript. The same is actually true for some taxonomic terms (e.g., "caecilians" vs. "Caecilians"), too.

Lines 84–86: I agree with the authors that this is the best available dataset to use for this analysis. I am wondering if there are any biases (e.g., taxonomic or geographic) in the missing data. That is, how are the 17% of all amphibian taxa that are missing from this tree distributed across amphibian families, and how are they distributed across the globe? Might this potential non-random missing data create a bias in estimates of speciation rates (e.g., Pybus & Harvey 2000)? I think that some acknowledgment and consideration of this problem would strengthen the authors' interpretations.

Lines 101–102: This means that these grid cells are not equal in area across the globe. Is this difference not large enough for the authors to think it would influence analyses?

Lines 104–106: Here again, I am wondering what the biases are in the missing data. Are the ~25% of amphibians missing these distribution data randomly distributed with respect to phylogeny and geography? If not, do the authors think this could alter the interpretation of their results? Some consideration of this would be helpful.

Lines 126–128: A problem with this metric of heterogeneity is that it is expected to be biased by sample size (i.e., the number of species within each group). Even if speciation rates are random, we would expect to measure greater heterogeneity (by this measure) in grid-cells with more species—potentially creating a spurious correlation that is really driven by geographic differences in diversity. The same is not true for other measures of variance (e.g., standard deviation), so I'm not sure why the authors opted to measure heterogeneity in this way. I encourage the authors to consider these alternatives and/or provide a more thorough justification for the use of this measure. I think that this is especially necessary because of the importance the authors place upon this observation in their broader explanations for how to reconcile a LDG and inverse latitudinal speciation gradient (e.g., Lines 304–313; Lines 399–400).

Lines 222–226: This seems to support the concern that this metric of variance may be biased by diversity (see comments on Lines 126–128).

Line 275: Did the authors fit a single model with multiple independent variables (i.e., what Table 3 seems to suggest), or did the authors fit a bunch of separate models, each with a single independent variable (i.e., what Figure 4 seems to suggest)? Inclusion of the code with the submission would help reviewers determine this.

Figure 4: The point in the upper right corner of the “topographic complexity” plot seems like it could be driving this relationship. I suggest that the authors address this to evaluate whether this pattern is truly robust (e.g., through a test for leverage).

Lines 344–347: If I understand correctly, this probably doesn’t need to be speculative. If they stick to the metric of heterogeneity they use here, the authors could actually calculate the contribution of caecilians to these estimates (e.g., what proportion of areas have the lower end of their speciation rate range set by a caecilian?).

Lines 352–353: I agree with the authors that high topographic complexity should create vicariate barriers and drive allopatric speciation. But in just the previous paragraph, they claim that “low dispersal capacity” (Line 343) should be associated with lower diversification. If the dispersal ability and dispersal barriers can be equally valuable in explaining high and low diversification rates, how valuable are they as a falsifiable hypothesis to test? I understand that these phenomena are complex and may have counteracting effects, but the authors may benefit from addressing this more directly.

Lines 371–373: This is true, but isn’t the method the authors used weighted towards more recent speciation events?

Reviewer #2

(Remarks to the Author)

This paper examines the latitudinal diversity gradient in amphibians by estimating speciation rates and examining various potential ecological factors that may drive spatial patterns of diversification in this group. The authors find support for a general inverse latitudinal gradient of higher speciation rates in higher latitudes and more mixed speciation rates at lower latitudes.

Overall the paper was excellent, and the authors have clearly laid out potential drivers of latitudinal variation of speciation rate and how these predictions could be testing using different patterns of data available.

l86: It is not clear whether the “full” tree that includes taxa placed using stochastic polytomy resolution from the Jetz paper was used to estimate DR. The authors should make this clearer if that is the case. If not, it would be inappropriate to use the DR statistic as the DR statistic requires a complete phylogeny, as it decreases the weight of branches deeper (more ancient) in the phylogeny based on the number of intervening nodes.

l96: Vasconcelos et al 2022 actually suggests, in contrast to Title and Rabosky 2019, that DR performs poorly relative to model-based approaches such as BAMM, CLADS and MiSSE. Could the authors add something to the discussion that talks about this point and how some of the scenarios for DR in that paper could impact their results? For example, Vasconcelos consistently found that DR struggled to estimate diversity-dependent diversification, which could explain why NRI was not found to be a significant predictor of speciation rates.

l364: Can you make this point a little bit clearer? I think what is trying to be said here is something along the lines of, lower extinction rates would permit relict taxa to survive to the present due to climactic stability, but because we observe higher speciation rates in unstable regions, we don’t believe that to be case for amphibians. It’s not clear to me how the two relate to each other as the DR method used in this paper can only really estimate speciation rates, so any discussion of drivers of extinction are inappropriate. Possibly one would expect high turnover to correlate with high rates of speciation only, but I think it is a bit speculative to make that argumentative leap here.

https://tncvasconcelos.github.io/papers/Vasconcelos_et_al_2022_misse.pdf
<https://besjournals.onlinelibrary.wiley.com/doi/10.1111/2041-210X.13153>

Version 1:

Reviewer comments:

Reviewer #1

(Remarks to the Author)

In my first review of this manuscript, I thought that it was well-conceived and executed, but I had a few questions about potential biases (e.g., due to missing data, methods use in calculating variance, etc.). Although some of those questions remain, I think that the authors have made strong efforts to address these potential challenges and acknowledged them in the manuscript. I think this work should be published, and I have no further recommendations for revisions.

Reviewer #2

(Remarks to the Author)

I have read the revised manuscript and the authors' response letter. I am satisfied that the manuscript is significantly improved and much clearer in its conclusions.

Reviewers' comments:

Reviewer #1 (Remarks to the Author):

The authors provide an analysis of global speciation rates in amphibians—demonstrating that the latitudinal diversity gradient is not simply explained by higher speciation rates in the tropics. In fact, speciation rates (overall, which was driven by frogs) are higher at higher absolute latitudes, and the authors provide some additional analyses to attempt to disentangle potential explanations. Overall, I found the study to be well-conceived and the manuscript to be well-written and clear. I think that it's a valuable contribution to the field. Below, I provide some comments—the most important of which I believe to be related to the implications of the way the authors calculated heterogeneity. Barring that, I have no major concerns or suggestions for improvement.

R/ We appreciate the reviewer's overall positive and constructive criticism. Below we explain how we addressed the reviewer's concerns.

Lines 11–12: Where exactly does the claim that "we detected peaks of speciation in both low- and high-latitude taxa" come from? To me, this implied that the authors would document and discuss taxon-specific speciation rates somewhere in the paper, but I couldn't find these data in the results or supplemental materials (although there is some discussion of the bolitoglossine plethodontids from Lines 329–335). If this is not what they mean to imply, perhaps this could be rephrased for clarity or removed from the abstract, where I think it is given greater emphasis than elsewhere in the manuscript.

R/ Thanks for the observation, we have rephrased the sentence to make our point clearer. The related text now reads as: "Despite the overall inverse latitudinal trend in mean speciation rates of amphibians, we found that tip-level maximum speciation rates are not necessarily restricted to higher latitudes and can be found in different regions across the globe". (Lines 12-14).

Also, as explained below, we modified Figure 2 to show the distribution of per-grid maximum speciation values and rate heterogeneity based on the coefficient of variation of speciation rates. We think that the description and discussion of these results support a pattern of regional variation and also better contextualize the final statement of the abstract. Regarding this, we enriched the discussion in the following paragraph where we provide taxon-specific examples:

"In amphibians, we also found an inverse latitudinal speciation gradient relative to species richness. Nevertheless, we also noted that speciation rates may vary highly within species assemblages in some regions of the world. This may have an important -yet underestimated- impact on calculating mean speciation values, on which descriptions of latitudinal gradients are often based. Our results show that co-occurring fast and slow speciating taxa is widespread in species-rich assemblages, such as those in tropical regions (e.g. Isthmian Central America, Madagascar, the Amazon basin, or Southeast Asia) or regional hotspots of amphibian diversity in other latitudes (e.g. southeastern USA, (Jenkins et al., 2015)). For example, in the Mesoamerican hotspot the distribution of the fast-speciating tropical Bolitoglossine salamanders overlaps with other species showing very slow rates of speciation, such as the Mexican burrowing toad, *Rhynophrynus dorsalis*. In many cases, these co-occurrence patterns translate into intermediate mean speciation rates in species rich assemblages at lower latitudes, contrary to several high latitude regions, with less diverse amphibian faunas but with estimated speciation rates above the average. For example, frogs of the genus *Alsodes* and plethodontid salamanders of various genera (e.g. *Eurycea* and *Plethodon*) are distributed out of the Tropics, respectively, and show some of the fastest speciation rates among the over 7200 species studied. Certainly, the co-occurrence of fast and slow speciating lineages in some regions reveals the importance of accounting for the idiosyncratic responses of lineages to shared eco-evolutionary pressures, and which could lead to fast/slow speciation in some groups but not in others.

Likewise, other sources of rate heterogeneity such as the variation in speciation rates among groups diversifying at different times should be better incorporated to fully understand the temporal dynamics that result in traditional – following species richness – as well as inverse – opposite to species richness – latitudinal gradients of speciation (Stephens et al., 2025).” (Lines 347-370).

Line 40: A minor point here. At various points, the authors either do or don't capitalize “tropics” (and “tropical”). I encourage them to format this consistently throughout the manuscript. The same is actually true for some taxonomic terms (e.g., “caecilians” vs. “Caecilians”), too.

R/ Thanks for spotting this. We have now standardized the terms throughout the manuscript using non-capitalized style.

Lines 84–86: I agree with the authors that this is the best available dataset to use for this analysis. I am wondering if there are any biases (e.g., taxonomic or geographic) in the missing data. That is, how are the 17% of all amphibian taxa that are missing from this tree distributed across amphibian families, and how are they distributed across the globe? Might this potential non-random missing data create a bias in estimates of speciation rates (e.g., Pybus & Harvey 2000)? I think that some acknowledgment and consideration of this problem would strengthen the authors' interpretations.

R/ This is a very good point, we appreciate the suggestion as this could be particularly relevant for amphibians given the fast rate of species description in this group over the last decades. The phylogeny we used was published in 2018, nevertheless, it was based on sequences previously compiled and on the AmphibiaWeb taxonomic backbone as of 2014. Therefore, to assess the geographic and taxonomic bias in the missing data we compiled the full list of species described since then (2015-to date), which likely represent the larger portion of missing tips in the phylogeny.

We found that over the last decade, more than 1400 new amphibian species have been described. The majority are species belonging to the families Strabomantidae (186 spp), Microhylidae (178 spp), Megophryidae (130 spp), Hylidae (116 spp) and Ranidae (86 spp).

These species account for 49% of the newly described amphibian taxa. Nevertheless, the new descriptions for this period also include one or more species from 47 other families. Looking at the geographical pattern, large numbers of these new species have been described in China (192 spp), Brazil (175 spp), Madagascar (126 spp), Ecuador (113 spp), Peru (86 spp) and India (78).

We also mapped the distribution of the subset of species that are not included in the phylogeny, but only for those which range maps are available at IUCN. The resulting map also confirmed the bias of the missing data towards some of these regions, being the most evident in the Neotropical region.

We added text to the discussion, arguing the implications of this in the light of potential bias in our speciation estimates and included the reference you mentioned. The relevant sentences related to this issue complement the point previously highlighted in the discussion on the potential effect of the Latitudinal taxonomy gradient proposed by Freeman and Pennell (2021), and now read as:

“After a brief review assessing geographic and taxonomic biases in the species not included in the phylogeny here used, we found that among the c. 1,400 amphibian species described in the past decade, a significant portion are tropical, with nearly 50% clustered within just five of 52 families recently represented. Although incorporating proposed solutions to address these biases is challenging (Freeman & Pennell, 2021), acknowledging them is important due to their potential impact on speciation estimates (Pybus & Harvey, 2000). Notably, the exclusion of numerous tropical lineages may lead to an underestimation of dynamic diversification in these regions.” (Lines 441-448).

Lines 101–102: This means that these grid cells are not equal in area across the globe. Is this difference not large enough for the authors to think it would influence analyses?

R/ Given the approach used we consider that these differences should not have a significant impact on our analyses. While it is true that some variation in cell area is expected, this variation is mainly evident in higher latitudes where only a few species for some clades are distributed. Moreover, we used species' presence to derive both features of interest (richness and speciation), therefore we consider this would not be a major problem. This logic is also supported by evidence showing that using range maps/polygons to describe richness gradients at 1° grid-cells is reliable to derive robust patterns (Hurlbert & Jetz, 2007). Finally, and perhaps more importantly, the analyses involving predictor variables of speciation rate variation were done by considering bioregions as analytical units. Then, in this case, using non-projected polygons would not be a problem since we only used the bioregion "membership" of species to compute speciation rate values at that level (instead of values per grid-cell).

Lines 104–106: Here again, I am wondering what the biases are in the missing data. Are the ~25% of amphibians missing these distribution data randomly distributed with respect to phylogeny and geography? If not, do the authors think this could alter the interpretation of their results? Some consideration of this would be helpful.

R/ Thanks for the observation. As mentioned above, based on our evaluation of the geographic and taxonomic characteristics of the missing data we have now commented on this drawback in our revised discussion section. Please see the lines (433-440) that are highlighted above.

Lines 126–128: A problem with this metric of heterogeneity is that it is expected to be biased by sample size (i.e., the number of species within each group). Even if speciation rates are random, we would expect to measure greater heterogeneity (by this measure) in grid-cells with more species—potentially creating a spurious correlation that is really driven by geographic differences in diversity. The same is not true for other measures of variance (e.g., standard deviation), so I'm not sure why the authors opted to measure heterogeneity in this way. I encourage the authors to consider these alternatives and/or provide a more thorough justification for the use of this measure. I think that this is especially necessary because of the importance the authors place upon this observation in their broader explanations for how to reconcile a LDG and inverse latitudinal speciation gradient (e.g., Lines 304–313; Lines 399–400).

R/ Thanks for the suggestion, we fully agree on this. We have now modified Figure 2 to include a map showing the cell-by-cell coefficient of variation of speciation rates, which robustly assess variability while accounting for the effect of sample size or the scale of the data. This certainly is a better proxy to show the speciation rate heterogeneity we found in the different grid cells and how this could mask the existence of fast speciating groups at different latitudes, not only at high latitudes as perceived from simply looking at mean speciation rates. We have added new text explaining the inclusion of this metric: “Moreover, we created maps to visualize the distribution of maximum speciation rates estimated for each grid-cell and the distribution of speciation rate heterogeneity. For the latter, we calculated the coefficient of variation of speciation rates within grid-cells, a proxy of heterogeneity that is not prone to be influenced by differences in species richness among grid cells” (Lines 135-139).

Lines 222–226: This seems to support the concern that this metric of variance may be biased by diversity (see comments on Lines 126–128).

R/ Correct. We have included a different metric and modified the text in the new version to align with the narrative mentioned above. The whole paragraph now reads as: “Considering all species, we found that within grid cells maximum speciation rates, are not necessarily restricted to higher latitudes but occur in different regions across the globe, including eastern USA, Mesoamerica, the Amazon, the Atlantic Rainforest, Southern Andes, Madagascar, and Southeast Asia (Fig. 2). We found that in these regions the speciation rates are very heterogeneous, as inferred from the high grid cell level coefficient of variation in this metric, which in most cases exceeds 30% (Fig. 2). In contrast, we identified some of the most homogeneous grid cells in terms of speciation rates in the northern portions of both the Nearctic and Palearctic regions.” (Lines 246-253).

Line 275: Did the authors fit a single model with multiple independent variables (i.e., what Table 3 seems to suggest), or did the authors fit a bunch of separate models, each with a single independent variable (i.e., what Figure 4 seems to suggest)? Inclusion of the code with the submission would help reviewers determine this.

R/ We appreciate the observation. We did fit a single SAR model with multiple independent variables. We now added further information on our modeling approach in the methods section. Accordingly, the results shown in the original (as well as in the now modified) Figure 4 come from the single Spatial Autoregressive Model using all variables. The revised text now reads: “We tested for significant relationships among these predictor variables and the variation in mean speciation rates across bioregions by running additional SAR models with all predictors to account for spatial autocorrelation. Given the large areas covered by our study units, we determined maximum neighbor distances by fitting two sets of lag and error models, with maximum connectivity distances ranging from 10 to 100 degrees. From these models, we selected 40 degrees as the optimal distance based on the lowest Akaike

Information Criterion (AIC) value among all models. To estimate variance explained by the predictors while accounting for spatial structure, we calculated pseudo-R² based on the Nagelkerke formula (59).” (Lines 210-219).

Moreover, As the reviewer suggested, we fully support making the codes available. With this resubmission, we provide the corresponding access link.

Lines 344–347: If I understand correctly, this probably doesn’t need to be speculative. If they stick to the metric of heterogeneity they use here, the authors could actually calculate the contribution of caecilians to these estimates (e.g., what proportion of areas have the lower end of their speciation rate range set by a caecilian?).

R/ Thanks for the observation, indeed that is the case. We rephrased the statement to stress the importance of this group in contributing to within grid cell rate heterogeneity across its distribution by adding speciation values to the lower end of the speciation range of each grid cell.

The new text reads as: “The combination of this particular speciation dynamics in caecilians and their distribution mostly restricted to low latitudes, certainly contributes to the overall speciation rate heterogeneity found in some neotropical regions, where fast and slow speciation groups co-occur.” (Lines 402-405).

Figure 4: The point in the upper right corner of the “topographic complexity” plot seems like it could be driving this relationship. I suggest that the authors address this to evaluate whether this pattern is truly robust (e.g., through a test for leverage).

R/ Checking our data, and following the reviewer’s suggestion, we conducted a test of leverage and found that this upper right corner point is not an outlier.

In the plot above dashed lines represent the cutoff values to define outliers and were calculated as 2 and 3 times the proportion between number of variables and sampling points (i.e. $2*(p/n)$ and $3*(p/n)$). As seen, the point corresponds to the South American temperate bioregion. Upon further investigation, we found that this region is home to several species of the genus *Alsodes*, which despite not being highly diverse ($n= 20$ spp), happens to be one of the clades with the highest speciation rate estimations in our dataset. Given the results of the leverage test, and the explanation we found for this particular region, we decided to retain the original model as it was.

Lines 352–353: I agree with the authors that high topographic complexity should create vicariate barriers and drive allopatric speciation. But in just the previous paragraph, they claim that “low dispersal capacity” (Line 343) should be associated with lower diversification. If the dispersal ability and dispersal barriers can be equally valuable in explaining high and low diversification rates, how valuable are they as a falsifiable hypothesis to test? I understand that these phenomena are complex and may have counteracting effects, but the authors may benefit from addressing this more directly. Thank you for pointing out this apparent contradiction. The key distinction is that low dispersal can influence diversification rates differently depending on the ecological and evolutionary context of a given group. In most vertebrate clades, reduced dispersal can promote allopatric speciation by increasing population isolation. However, in caecilians, their extreme burrowing specialization appears to have restricted their range expansion to the point that it limits opportunities for diversification rather than promoting it. This contrasts with groups like anurans and salamanders, where limited dispersal in topographically complex regions has been associated with higher speciation rates. We have now added text trying to clarify this point:

“This confirms previous findings suggesting that the specialized and evolutionary conserved ecology of caecilians along with their low dispersal capacity may have constraint their diversification (Pyron, 2014; Wiens, 2007; Zhang & Wake, 2009), in contrast to other groups where reduced dispersal has been linked to increased speciation rates (Ikeda et al., 2012; Schluter & Conte, 2009). The combination of this particular speciation dynamics in caecilians and their distribution mostly restricted to low latitudes, certainly contributes to the overall speciation rate heterogeneity found in some neotropical regions, where fast and slow speciation groups co-occur.” (Lines 398-405).

Lines 371–373: This is true, but isn’t the method the authors used weighted towards more recent speciation events?

R/ Thanks for the observation, you are right. We used phylogenetic methods (tip. rates) to capture speciation events, which tend to match much better with the available paleoclimatic datasets. However, more profound evolutionary events, which span 300 Myrs, are not captured by our spatially explicit approach and likely require other computational approaches and datasets (e.g., fossils) to answer questions related to latitudinal variation of speciation-extinction rates.

Reviewer #2 (Remarks to the Author):

This paper examines the latitudinal diversity gradient in amphibians by estimating speciation rates and examining various potential ecological factors that may drive spatial patterns of diversification in this group. The authors find support for a general inverse latitudinal gradient of higher speciation rates in higher latitudes and more mixed speciation rates at lower latitudes. Overall the paper was excellent, and

the authors have clearly laid out potential drivers of latitudinal variation of speciation rate and how these predictions could be testing using different patterns of data available.

R/ We truly appreciate the very positive and encouraging feedback on our manuscript. Below, we provide the details on how we addressed the points here raised.

l86: It is not clear whether the "full" tree that includes taxa placed using stochastic polytomy resolution from the Jetz paper was used to estimate DR. The authors should make this clearer if that is the case. If not, it would be inappropriate to use the DR statistic as the DR statistic requires a complete phylogeny, as it decreases the weight of branches deeper (more ancient) in the phylogeny based on the number of intervening nodes.

R/ Yes, you are correct, we used the full trees with the imputed species. In lines 88-89 this is mentioned as: "We estimated tip-level speciation rates across the most complete amphibian phylogeny containing 7238 species". (Lines 88-90).

To make it clearer, in this version, we have specified that the metric was estimated "from 100 random full trees (i.e. containing all 7238 species available) ...". (Line 103).

l96: Vasconcelos et al 2022 actually suggests, in contrast to Title and Rabosky 2019, that DR performs poorly relative to model-based approaches such as BAMM, CLADS and MiSSE. Could the authors add something to the discussion that talks about this point and how some of the scenarios for DR in that paper could impact their results? For example, Vasconcelos consistently found that DR struggled to estimate diversity-dependent diversification, which could explain why NRI was not found to be a significant predictor of speciation rates.

R/ Thanks for this valuable observation. We initially decided to use the DR metric because its calculation is computationally less demanding and, thus, much faster than obtaining estimations from model-based approaches and our phylogeny is quite large. Indeed, conducting BAMM across 100 trees from the posterior with >7000 species each requires enormous computational processing time. Still, we understand and share the reviewer's concern. Therefore, for this revised version, we conducted ten BAMM and ten ClADS runs on the same number of random trees from the posterior distribution. Then, we used mean speciation values resulting from each model to explore the correlation among the three metrics, both at the tip (species) and at the assemblage level. In all comparisons, we found strong correlations ($r > 0.7$) between metrics.

This suggests that the overall patterns is robust and will not change depending on the metric as, on average, species/assemblages with higher (or slower) rates for one metric would still present higher (or lower) rates with any other metric; especially in the case of assemblage where the value is pooled across species. As such, we opted to keep our initial DR estimates from 100 trees and subsequent analyses using them. This is now explained in the text as:

“We explored the consistency of our estimates with those of model-based metrics by implementing BAMM and ClaDS analyses, but only on a smaller set of ten random trees. Given the strong correlation between the average speciation rates obtained with the three approaches, for both species and assemblages, ($r > 0.7$; see Fig. S1), we opted to use DR for subsequent analyses to better account for topological uncertainty using 100 instead of ten trees.” (Lines 104 - 109).

Still, we appreciate your suggestion and consider it appropriate, therefore we also added new text to the discussion on this point:

“Finally, our chosen speciation rate metric has been criticized as performing poorly in some scenarios, compared to model-based metrics such as those derived from BAMM or ClaDS (Vasconcelos et al., 2022). However, our exploratory analyses showed that tip-level (species) and assemblage-level values of speciation rate are highly correlated between these three metrics ensuring that the overall patterns will not change depending on the metric as, on average, species/assemblages with higher (or slower) rates for one metric would still present higher (or lower) rates with any other metric; especially in the case of assemblage where the value is summarized across species.” Lines (451-457)

I364: Can you make this point a little bit clearer? I think what is trying to be said here is something along the lines of, lower extinction rates would permit relict taxa to survive to the present due to climactic

stability, but because we observe higher speciation rates in unstable regions, we don't believe that to be the case for amphibians. It's not clear to me how the two relate to each other as the DR method used in this paper can only really estimate speciation rates, so any discussion of drivers of extinction are inappropriate. Possibly one would expect high turnover to correlate with high rates of speciation only, but I think it is a bit speculative to make that argumentative leap here.

R/ Thank you for highlighting this point. We agree that the original text could more clearly separate the potential influences of climatic stability on extinction rates from our focus on speciation rates. We acknowledge that the DR metric we used is suited for estimating speciation rates and does not account for extinction processes. Thus, while climatic stability may theoretically allow relict taxa to persist, our results indicate that regions with higher climatic instability are associated with higher speciation rates, which suggests that, for amphibians, this instability may have driven speciation rather than merely permitting survival. We have revised the text to clarify this distinction and avoid speculative conclusions about extinction.

The new paragraph reads as: “Long-term regional climatic stability has often been proposed as a driver of biodiversity potentially reducing extinction rates and allowing taxa to persist in stable environments (Willig et al., 2003). Stability might also enable time for morphological differentiation and promote specialization that could, in theory, enhance speciation rates (Fine, 2015). However, our results reveal an increase in mean speciation rates in regions that have experienced greater climatic instability over the past ~3.3 million years (Mya). This aligns with findings in other groups, such as mammals, where higher speciation rates are linked to regions of climatic instability, measured as the difference in mean annual temperature between the present and the last glacial maximum (Morales-Barbero et al., 2020), a more recent timeframe than in our study. Given the evolutionary timeline of amphibians (ca. 300 Mya), we chose to use the longest available climatic dataset to capture more extended periods of environmental change. While we recognize that only a portion of the studied species are products of speciation within the last ~3 Mya, these results suggest that regions of climatic instability may indeed foster speciation in amphibians, despite their deep evolutionary history. We stress that the ability of past climatic dynamics to inform about processes on much deeper evolutionary scales must be interpreted cautiously and remain an issue to be improved in future studies”. **(Lines 417-432).**

References mentioned in this response to reviewers

- Fine, P. V. A. (2015). Ecological and evolutionary drivers of geographic variation in species diversity. *Annual Review of Ecology, Evolution, and Systematics*, 46, 369–392. <https://doi.org/10.1146/annurev-ecolsys-112414-054102>
- Freeman, B. G., & Pennell, M. W. (2021). The latitudinal taxonomy gradient. *Trends in Ecology and Evolution*, 36(9), 778–786. <https://doi.org/10.1016/j.tree.2021.05.003>
- Hurlbert, A. H., & Jetz, W. (2007). Species richness, hotspots, and the scale dependence of range maps in ecology and conservation. *Proceedings of the National Academy of Sciences of the United States of America*, 104, 13384–13389. <https://doi.org/10.1073/pnas.0704469104>
- Ikeda, H., Nishikawa, M., & Sota, T. (2012). Loss of flight promotes beetle diversification. *Nature Communications*, 3. <https://doi.org/10.1038/ncomms1659>
- Jenkins, C. N., Van Houtan, K. S., Pimm, S. L., & Sexton, J. O. (2015). US protected lands mismatch biodiversity priorities. *Proceedings of the National Academy of Sciences of the United States of America*, 112(16), 5081–5086. <https://doi.org/10.1073/pnas.1418034112>
- Morales-Barbero, J., Gouveia, S. F., & Martinez, P. A. (2020). Climatic instability predicts the inverse latitudinal pattern in net diversification of modern mammalian biota. *Journal of Evolutionary Biology*. <https://doi.org/10.1111/jeb.13737>
- Pybus, O. G., & Harvey, P. H. (2000). Testing macro-evolutionary models using incomplete molecular phylogenies. *Proceedings of the Royal Society B: Biological Sciences*, 267(1459), 2267–2272. <https://doi.org/10.1098/rspb.2000.1278>
- Pyron, R. A. (2014). Biogeographic analysis reveals ancient continental vicariance and recent oceanic dispersal in amphibians. *Systematic Biology*, 63(5), 779–797. <https://doi.org/10.1093/sysbio/syu042>
- Schluter, D., & Conte, G. L. (2009). Genetics and ecological speciation. *Proceedings of the National Academy of Sciences*, 106, 9955–9962. www.nasonline.org/SacklerDarwin.
- Stephens, P. R., Farrell, M. J., Davies, T. J., Gittleman, J. L., Meiri, S., Moreira, M. O., Roll, U., & Wiens, J. J. (2025). Global Diversity patterns are explained by diversification rates and dispersal at ancient, not shallow, timescales. *Systematic Biology*. <https://doi.org/10.1093/sysbio/syaf018/8071383>
- Vasconcelos, T., O'Meara, B. C., & Beaulieu, J. M. (2022). A flexible method for estimating tip diversification rates across a range of speciation and extinction scenarios. *Evolution*, 76(7), 1420–1433. <https://doi.org/10.1111/evo.14517>
- Wiens, J. J. (2007). Global patterns of diversification and species richness in amphibians. *The American Naturalist*, 170(March), 86–106. <https://doi.org/10.1086/519396>
- Willig, M. R., Kaufman, D. M., & Stevens, R. D. (2003). Latitudinal gradients of biodiversity : pattern, process, scale, and synthesis. *Annual Review of Ecology and Systematics*, 34, 273–309. <https://doi.org/10.1146/annurev.ecolsys.34.012103.144032>

Zhang, P., & Wake, M. H. (2009). A mitogenomic perspective on the phylogeny and biogeography of living caecilians (Amphibia: Gymnophiona). *Molecular Phylogenetics and Evolution*, 53(2), 479–491. <https://doi.org/10.1016/j.ympev.2009.06.018>